# DISSECTING IMPLICIT CHAIN OF THOUGHT: CAN TRANSFORMERS LEARN IT SPONTANEOUSLY?

## ABSTRACT

Large language models have gained powerful reasoning abilities through step-by-step reasoning chains, enabling deep logical reasoning and complex task handling. However, the shift from fast thinking to *slow thinking* (i.e., lengthy explicit reasoning steps) leads to massive token consumption, severely compromising practical reasoning efficiency. To address this, existing studies attempt to compress reasoning chains into latent tokens (Implicit CoT), but they often suffer from significant performance loss due to inadequate expression of original reasoning semantics. This paper theoretically analyzes the additional training costs of Implicit CoT when latent tokens lack effective supervision: as more steps are compressed, the originally efficient learning chain learning degrades exponentially, even reverting to the performance of "no CoT" when all intermediate processes are compressed. To solve this, we propose a distribution alignment method that adds moderate supervisory information to guide latent token distribution. Experimental results show that intermediate state supervision effectively improves the learning efficiency and stability of implicit CoT, significantly mitigating its reasoning performance decline.

## 1 INTRODUCTION

Large language models (LLMs) have demonstrated strong performance on complex reasoning, particularly under the Chain of Thought (CoT) paradigm (Wei et al., 2022), which decomposes intricate reasoning problems into sequential steps to enhance reasoning ability (Zhao et al., 2023; Lu et al., 2022; Hendrycks et al., 2021). Recently, *slow thinking* approaches (Li et al., 2025) such as GPT4-o1 (OpenAI, 2025) and DeepSeek (Guo et al., 2025) have gained attention for their effectiveness in mathematical and commonsense reasoning (Zhao et al., 2023; Ahn et al., 2024). However, these methods typically produce lengthy intermediate reasoning steps, which incur substantial computational costs (Zhu et al., 2025b; Zhang et al., 2023b). Furthermore, studies suggest that these approaches suffer from verbosity issue. For example, Warner et al. (2025) showed that many tokens generated during "thinking" process primarily serve linguistic fluency rather than substantive reasoning. To address this issue, several efforts have proposed guiding models to produce more concise reasoning chain, thereby reducing unnecessary linguistic redundancy (Warner et al., 2025; Yan et al., 2025). While these methods have optimized linguistic conciseness to some extent, they remain limited within the natural language space and cannot fundamentally replace the complete reasoning process (Hao et al., 2024).

To further explore the reasoning capabilities of large language models in a broader potential space, existing research has begun to attempt transforming the explicit discrete token representation of natural language reasoning chains into dense, continuous representations (Implicit CoT and Continuous Thoughts) (Xu et al., 2025; Shen et al., 2025; Hao et al., 2024). A typical approach, as illustrated in Figure 1(b), implicit reasoning chains use special latent vectors to replace reasoning steps (Zhang et al., 2025b), which not only aligns better with the compactness and abstraction of human reasoning processes but also frees itself from the constraints of the natural language space, providing the model with a more expressive potential representation. These studies offer new possibilities for enhancing reasoning efficiency and performance while also opening new perspectives for understanding the underlying reasoning mechanisms of language models.

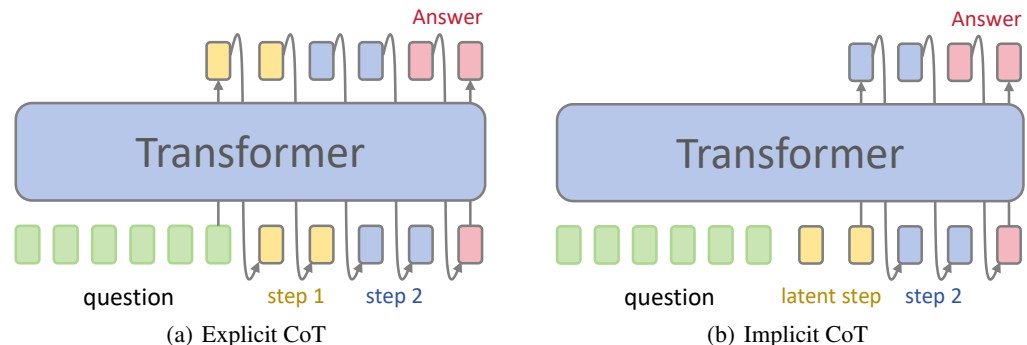

Figure 1: Comparison of explicit and implicit Chain-of-Thought paradigms. Our paper specifically analyses the implicit approach, in which latent tokens serve as intermediate reasoning steps. These methods insert special tokens into the CoT context to capture the reasoning semantics of the original steps. During the reasoning phase, this enables faster inference.

However, while implicit reasoning chains (Implicit CoT) offer a new research direction for improving reasoning efficiency, existing methods generally face the challenge of declining reasoning performance, making it difficult to match the impressive results of explicit reasoning chains (CoT) in complex reasoning tasks (Zhu et al., 2025a; Hao et al., 2024). This performance loss poses a significant challenge to the practical application and feasibility of implicit reasoning chains.

To elucidate the limitations of implicit reasoning chains in solving reasoning tasks, this study systematically investigates the inefficiencies of implicit CoT reasoning—implemented via the introduction of special latent "thinking" tokens—through an analysis of the parity problem. Our theoretical results identify two fundamental drawbacks of current implicit CoT approaches: (i) latent tokens in implicit reasoning chains fail to effectively encode intermediate reasoning processes; and (ii) although implicit reasoning chains can eventually acquire problem-solving patterns after extensive training, their learning difficulty grows exponentially with the number of compressed steps. These findings underscore the necessity of providing effective supervision for latent tokens in implicit reasoning chains, a conclusion further corroborated by our experimental results on the parity problem.

Our main contributions are as follows:

• **Reasoning ability of implicit CoT under unsupervised signals**: We theoretically reveal the drawbacks of the current implicit CoT in efficiently learning reasoning abilities. Our conclusion points out that latent tokens struggle to effectively learn intermediate reasoning processes autonomously.

• **Training costs brought by implicit CoT**: We theoretically analyze the additional training costs incurred when compressing the reasoning chain without effective supervision of latent tokens in implicit CoT. Our analysis indicates that as more steps are compressed, the originally efficient learning method of the reasoning chain degrades exponentially, even reverting to a level equivalent to not using a reasoning chain at all when all intermediate processes are compressed.

• **Distribution alignment method for training implicit CoT**: Based on the above findings, we believe that effective supervision of the potential tokens in implicit reasoning chains is necessary. We propose a distribution alignment method to guide the learning of latent tokens, thereby improving the learning efficiency of the model's implicit CoT. Our experimental results demonstrate that intermediate state supervision can effectively enhance the learning efficiency and stability of implicit reasoning chains, significantly alleviating the decline in implicit CoT reasoning performance.

## 2    RELATED WORK

Discrete tokens, serving as symbolic representations of intermediate reasoning steps or cognitive operations, have emerged as a promising paradigm to boost LLMs' reasoning capabilities, significantly improving task execution efficiency and performance. A popular reasoning paradigm in implicit Chain-of-Thought (Implicit CoT) involves replacing original reasoning steps with special latent

tokens to increase reasoning efficiency Zhu et al. (2025a); Chen et al. (2025). Prior to these studies, research had already explored the use of learnable "pause tokens" in prompts to assist model thinking Goyal et al. (2024). Additionally, Pfau et al. (2024) investigated the application of padding tokens in LLM reasoning, suggesting that they help models address parallelization issues.

Recent studies have found that by progressively shortening the CoT, it can be compressed into a limited number of tokens Hao et al. (2024), thereby improving reasoning efficiency. Shen et al. (2025) introduced a teacher model at the final answer of the model based on progressively compressing reasoning steps. By minimizing the difference between the teacher and student hidden activations, explicit CoT supervision is injected into the implicit CoT generation process. Recently, Zhang et al. (2025b) proposed a training framework for implicit CoT language models based on thinking tokens, promoting the model's rapid reasoning capabilities. Despite the potential of implicit reasoning chains in enhancing reasoning efficiency, their reasoning performance often struggles to match that of explicit reasoning chains Zhu et al. (2025a), severely limiting their application scenarios.

## 3 PRELIMINARIES AND SETUP

This section first defines the mathematical notations used in the paper, then systematically introduces the foundational problem (parity problem), core object (Implicit CoT), and model architecture (Transformer) required for subsequent analysis, laying the groundwork for theoretical and experimental discussions.

*Notation.* We denote the set $\{1, 2, \ldots, n\}$ as $[n]$. Vectors and matrices are denoted in bold text (e.g., $\boldsymbol{x}, \boldsymbol{A}$), whereas scalars appear in plain text (e.g., $y$). For $\boldsymbol{z} \in \mathbb{R}^n$ we write $\phi(\boldsymbol{z}) = (\phi(z_1), \cdots, \phi(z_n))^\top$, $\boldsymbol{z}^2 = \boldsymbol{z} \odot \boldsymbol{z} = (z_1^2, \cdots, z_n^2)$ and $|\boldsymbol{z}| = (|z_1|, \cdots, |z_n|)^\top$. The 2-norm is always denoted by $\|\cdot\|$. The multi-linear inner product or contraction of $\boldsymbol{z}_1, \cdots, \boldsymbol{z}_r \in \mathbb{R}^n$ for any $r \in \mathbb{N}$ is denoted as $\langle \boldsymbol{z}_1, \cdots, \boldsymbol{z}_r \rangle := \sum_{i=1}^n z_{1,i} \cdots z_{r,i}$. In particular, $\langle \boldsymbol{z}_1 \rangle = \boldsymbol{z}_1^\top \mathbf{1}_n$ and $\langle \boldsymbol{z}_1, \boldsymbol{z}_2 \rangle = \boldsymbol{z}_1^\top \boldsymbol{z}_2$.

### 3.1 THE PARITY PROBLEM

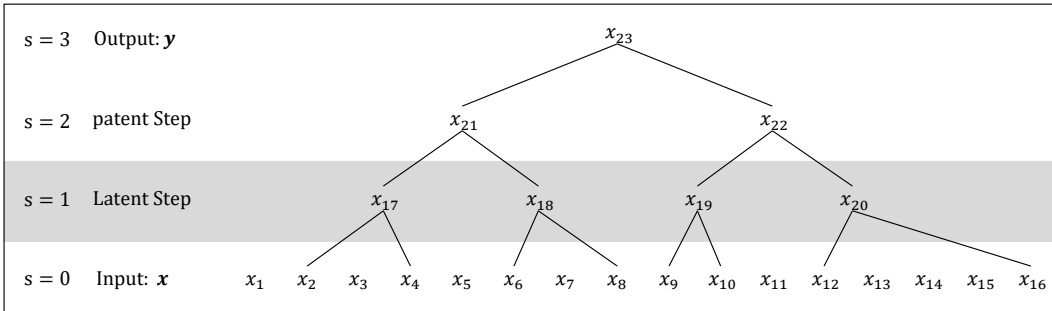

Figure 2: Schematic of the k-Parity Problem. $\boldsymbol{x}_{17}, \cdots, \boldsymbol{x}_{22}$ represent intermediate reasoning steps, which are replaced by latent tokens in the case of implicit Chain-of-Thought.

Given a set $\{1, \cdots, d\}$, for $d$-bit inputs $\boldsymbol{x} = (x_j)_{j=1}^d \sim \text{Unif}(\{\pm 1\}^d)$, the $k$-parity problem involves predicting $y = \prod_{j \in p} x_j$, where $p \subseteq [d]$ and $|p| = k$. In this problem, $p$ is unknown, and we denote the set of all possible $p$ as $P_k$, so $|P_k| = \binom{d}{k}$. Our problem setup follows the definition in Kim & Suzuki (2025) for theoretical analysis. We abuse notation and identify the set of indices $p$ with the corresponding parity mapping $\boldsymbol{x} \mapsto \prod_{j \in p} x_j$. Given $n$ samples $(\boldsymbol{x}^i, y^i)_{i \in [n]}$, our goal is to predict the parity of any test input.

Specifically, let $f_\theta : \{\pm 1\}^d \to \mathbb{R}$ be any differentiable parametrized model and suppose we select the target parity $p$ uniformly at random from $P_k$. In the finite-sample setting, $n$ i.i.d. samples $(\boldsymbol{x}^i, y^i)_{i \in [n]}$ are generated as $\boldsymbol{x}^i \sim \text{Unif}(\{\pm 1\}^d)$, $y^i = p(\boldsymbol{x}^i)$ and we are given access to (approximate) gradients from the empirical loss:

$$L_n^{\text{Dir}}(\theta) = \frac{1}{2n} \sum_{i=1}^{n} (y^i - f_\theta(\boldsymbol{x}^i))^2 = \frac{1}{2} \|p - f_\theta\|_n^2, \tag{1}$$

where $\|\cdot\|_n$ is the empirical norm. For each sample $\boldsymbol{x}$, $f_\theta$ acts independently, and the model must implicitly leverage correlations between samples in the average gradient to learn.

Another scenario arises when we consider training the model with CoT. As shown in Figure 2, we take 2 elements from the subset $p$ at a time, multiply them, and use the result as an intermediate state, which serves as new input for subsequent reasoning steps. In the case of CoT, the $k$-parity problem is decomposed into $h$ reasoning steps, with each step solving multiple 2-parity problems based on information from the previous step, until the final step outputs the solution $\boldsymbol{y}$ for the entire parity problem, where we denote $\mathsf{p}[\alpha]$ and $\mathsf{h}[\alpha]$ as the indices of the child and parent nodes of $\boldsymbol{x}_\alpha$, respectively. The training objective of the model then becomes minimizing the mean squared error loss at each step:

$$L_n^{\text{CoT}}(\theta) = \frac{1}{2n} \sum_{i=1}^{n} \sum_{j=d+1}^{d+k-1} (x_j^i - \hat{x}_j^i)^2 = \frac{1}{2n} \sum_{j=d+1}^{d+k-1} \|\boldsymbol{x}_j - \hat{\boldsymbol{x}}_j\|^2. \tag{2}$$

## 3.2 IMPLICIT CHAIN-OF-THOUGHT

A common approach to implicit Chain-of-Thought (Implicit CoT) is to gradually replace reasoning steps in the reasoning chain by adding "thinking tokens" (Zhang et al., 2025b; Hao et al., 2024), rather than abandoning the chain structure entirely, aiming to compress reasoning steps into a limited number of latent tokens.

In the implicit reasoning chain of the parity problem, we consider hiding an entire layer of the solution tree at a time. As shown in Figure 2, when Step 1 becomes an implicit reasoning chain, the sequence $[\boldsymbol{x}_{17}, \cdots, \boldsymbol{x}_{20}]$ is transformed from the original explicit reasoning steps into special latent tokens $c$. We maintain the number of latent tokens consistent with the number of tokens required for the original steps. Thus, each implicit token still only needs to handle a 2-bit parity problem. For ease of notation, we denote $\boldsymbol{c}^s = \{\boldsymbol{c}_j | j \in \{d + \tau_{s-1} + 1, \cdots, d + \tau_s\}\}$ as the set of latent tokens at step $s$, where $\tau_s$ is the total number of tokens up to step $s$ and $\boldsymbol{c}_j = (c_j^i)_{i=1}^n$. The training objective for each step of the implicit reasoning chain becomes predicting the subsequent intermediate states given the input $\boldsymbol{x}$ and latent tokens $\boldsymbol{c}^s$:

$$L_n^{\text{iCoT}}(\theta) = \frac{1}{2n} \sum_{i=1}^{n} \sum_{j=d+\tau_s+1}^{d+k-1} \left(x_j^i - \hat{x}_j^i\right)^2, \quad \hat{x}_j^i = f_\theta(x^i, \mathbf{c}^s, \hat{x}_{d+\tau_s+1}^i, \cdots, \hat{x}_{j-1}^i) \tag{3}$$

In an ideal scenario, to facilitate the learning of latent tokens, we would prefer to initialize $\mathbf{c}_s$ to the mean of the reasoning process's value range, allowing it to learn towards any value while avoiding significant impacts on the data distribution, i.e., $c_j^i = 0$ for all $i \in [n]$ and $j \in \{d + \tau_{s-1} + 1, \cdots, d + \tau_s\}$. However, initializing latent tokens to 0 can lead to gradient vanishing during actual training. Therefore, in our theoretical analysis, we set $c_j^i = \upsilon, \upsilon \to \mathbb{E}(x^i)$.[1]

## 3.3 TRANSFORMER MODEL

We discuss the learning ability of models in constructing implicit reasoning chains under a one-layer Transformer architecture. To simplify the analysis, we use absolute position encoding and single-head Softmax attention, and the single-layer Transformer also omits residual connections.

**Data encoding**: For the input $\boldsymbol{x}$, each input token $\boldsymbol{x}_j = (x_j^i)_{i=1}^n$, $j \in [d]$, is an $n$-dimensional vector, with its elements consisting of the $j$-th bit from $n$ samples. This differs slightly from the existing

---

[1]This setting is limited to the theoretical analysis process. In our subsequent experiments, we use one-hot encoding to randomly initialize latent tokens.

Transformer encoding method but is essentially the same [2]. For the latent tokens of the implicit reasoning chain, we initialize them to $\upsilon$ to facilitate the model's learning of intermediate hidden reasoning processes.

**Softmax attention layer**: The attention layer consists of key, query, and value matrices $\mathbf{K}, \mathbf{Q}, \mathbf{V}$. To make the dynamical analysis tractable, following Zhang et al. (2023a); Huang et al. (2023); Mahankali et al. (2023); Kim & Suzuki (2024), we focus on position encoding when obtaining attention scores, while the value matrix only focuses on $\boldsymbol{x}$. Therefore, in this paper, we run the position encoding $\boldsymbol{p}$ separately from the input $\boldsymbol{x}$ rather than concatenating them together. The forward propagation of the attention layer is defined as follows:

$$\hat{\boldsymbol{z}}_m = \sum_{j=1}^{m-1} \mathbf{V}\hat{\boldsymbol{x}}_j \cdot \mathsf{softmax}(\hat{\boldsymbol{p}}_j^\top \mathbf{K}^\top \mathbf{Q}\hat{\boldsymbol{p}}_m) = \sum_{j=1}^{m-1} \sigma_j(\boldsymbol{w}_m)\boldsymbol{x}_j, \tag{4}$$

**Feedforward layer**: To fully adapt to the parity problem, following Kim & Suzuki (2025), we use a mapping function $\phi : [-1, 1] \rightarrow [-1, 1]$, requiring $\phi(0) = -1, \phi(\pm 1) = 1$, and set the function to be smooth and differentiable. Finally, we define $\phi(t) = -1 + ct^2 + O(|t|^4)$ and $\phi'(t) = 2ct + O(|t|^3)$.

Thus, the transformer computes $\mathrm{TF}(\boldsymbol{x}_1, \cdots, \boldsymbol{x}_{d+k-1}; \mathbf{W}) = (\hat{\boldsymbol{x}}_1, \cdots, \hat{\boldsymbol{x}}_{d+k-1})$ where the original data $\hat{\boldsymbol{x}}_j = \boldsymbol{x}_j, j \in [d]$ remain unchanged and tokens $\hat{\boldsymbol{x}}_{d+1}, \cdots, \hat{\boldsymbol{x}}_{d+k-1}$ are computed as $\hat{\boldsymbol{x}}_m = \phi(\hat{\boldsymbol{z}}_m)$.

## 4 MAIN RESULTS

Based on theoretical analysis of the parity problem, this section systematically explores the limitations of Implicit CoT in learning reasoning processes. Our theoretical analysis based on the parity problem reveals two main challenges faced by implicit reasoning chains in learning reasoning knowledge. First, we prove that the "think token" in implicit reasoning chains struggles to effectively learn intermediate reasoning processes, limiting its ability to enhance reasoning capabilities. Second, we find that although implicit reasoning chains can still learn problem-solving patterns after extensive training, the learning difficulty increases exponentially with the number of compressed steps. These findings provide an important theoretical basis for understanding the limitations of implicit reasoning chains and offer guidance for future improvements in implicit reasoning chain methods.

### 4.1 LATENT TOKENS STRUGGLE TO LEARN REASONING KNOWLEDGE

For the learning ability of implicit reasoning chains, the intuitive argument is to assess the training status of latent tokens by estimating the magnitude of $\frac{\partial L_n^{\mathrm{CoT}}}{\partial \boldsymbol{c}_s}(\mathbf{W})$ in evaluation scenarios. However, for learning samples $\boldsymbol{x}^i \sim \mathrm{Unif}(\{\pm 1\}^d)$, making $\mathbb{E}_{d+\tau_s+1 \leq j < d+k-1, \boldsymbol{x}}[(x_j^i - \hat{x}_j^i)^2]$ approach 0 requires further theoretical analysis based on the efficient learning of explicit reasoning chains to understand the limitations of implicit reasoning chains.

Specifically, we define a set of intermediate states $t = x_{j_1}, \cdots, x_{j_r}$ for $1 \leq j_1, \cdots, j_r \leq d + k - 1$ as *trivial* if $\prod_{x \in t} x \equiv 1$. Correspondingly, we define $I_{r,m}$ as the set of nontrivial index $r$-tuples less than $m$: $I_{r,m} = \{(j_1, \cdots, j_r) \mid 1 \leq j_1, \cdots, j_r \leq m - 1, x_{j_1} \cdots x_{j_r} \not\equiv 1\}$.

Based on the theoretical analysis of CoT in Kim & Suzuki (2025), when solving the parity problem, the presence of *trivial* allows sub-nodes to effectively learn 2-bit parity problems through gradient signals from parent nodes during the training process of CoT.

**Theorem 1** (Efficient CoT Learning). *Suppose $n = \Omega(d^{2+\epsilon})$ for $\epsilon > 0$, $d$ is sufficiently large and let $\widetilde{\nabla}$ be any $O(d^{-2-\epsilon/8})$-approximate gradient oracle.[3] Set initialization $\mathbf{W}^{(0)} = \mathbf{0}$ and learning rate $\eta = \Theta(d^{2+\epsilon/16})$. Then for any target parity $p \in P_k$, it holds with probability $1 - \exp(-d^{\epsilon/2})$ over random sampling that for any $1 \leq m \leq d + k - 1, 1 \leq j \leq m - 1$, the gradient of the loss function $L_n^{CoT}$ with respect to $w_{j,m}$ at $\mathbf{W}$ is given by:*

---

[2]This can be intuitively understood as each bit being encoded as a 1-dimensional vector

[3]In fact, we only require that each component of the gradient has error at most $O(d^{-2-\epsilon/8})$ for Theorems 1, 2, which follows since the $L_\infty$ error is bounded above by $L_2$.

$$\frac{\partial L_n^{CoT}}{\partial w_{j,m}}(\mathbf{W}) = -\frac{2c}{(m-1)^2}1_{\{\mathbf{p}[j]=m\}} + O(d^{-2-\epsilon/8}) \tag{5}$$

Clearly, the key to efficient learning in explicit reasoning chains lies in the transmission of gradient signals between child and parent nodes. This allows the model to quickly focus attention on the corresponding child nodes during the learning process. To facilitate further discussion on the reasoning capabilities of implicit reasoning chains, we categorize the reasoning steps of the parity problem into three types based on the presence of child nodes and whether they are hidden:

(1) Implicit reasoning step set $S_i$: The subset of reasoning steps that are concealed within the implicit reasoning chain—e.g., $[\mathbf{x}_{17}, \ldots, \mathbf{x}_{20}]$ in Step 1 of Figure 2. These steps are replaced by latent tokens (e.g., $\mathbf{c}_1$) and receive no supervision from the original reasoning steps during training.

(2) Implicitly dependent reasoning step set $S_d$: The subset of reasoning steps that remain explicit but whose prerequisite step is hidden—e.g., $[\mathbf{x}_{21}, \mathbf{x}_{22}]$ in Step 2. These steps still receive supervision from their original reasoning content during training, but the preceding step they depend on is concealed.

(3) Explicit reasoning steps set $S_e$: The subset of reasoning steps that are fully explicit, with neither the step itself nor its prerequisite hidden—e.g., $[\mathbf{x}_{23}]$ in Step 3. These steps are directly supervised by their original reasoning content during training, and their required previous step is also explicitly accessible.

During the training of latent tokens in the implicit reasoning chain, the gradient of the objective function $L_n^{\text{iCoT}}$ can be decomposed into the sum of gradient signals from nodes in $S_d$ and $S_e$ (nodes in $S_i$ do not provide any gradient signals). We define the process of Transformer reasoning for each node as $f_m^\circ(\boldsymbol{x}^i; \mathbf{W}) = \hat{x}_{m,i}$. When discussing the learning ability of latent tokens, we focus on the gradient of node $f_m^\circ$ for $m \in S_i \cap S_e$ with respect to implicit token $w_{j,m}$ for $j \in S_i$ during backpropagation. We have the following conclusion, which contrasts with Theorem 1:

**Theorem 2.** *Suppose $n = \Omega(d^{2+\epsilon})$ for $\epsilon > 0$, $d$ is sufficiently large and let $\widetilde{\nabla}$ be any $O(d^{-2-\epsilon/8})$-approximate gradient oracle. Set initialization $\mathbf{W}^{(0)} = \mathbf{0}$ and learning rate $\eta = \Theta(d^{2+\epsilon/16})$. Then for any target parity $p \in P_k$, it holds with probability $1 - \exp(-d^{\epsilon/2})$ over random sampling that for any $m \in S_i \cap S_e, j \in S_i$, the gradient of the loss function $L_n^{\text{iCoT}}$ with respect to $w_{j,m}$ at $\mathbf{W}$ is given by:*

$$\frac{\partial L_n^{iCoT}}{\partial w_{j,m}}(\mathbf{W}) = O(d^{-2-\epsilon/8}) \tag{6}$$

Obviously, in implicit reasoning chains, the gradients returned by latent tokens make it difficult to effectively learn reasoning knowledge. On one hand, they decay at a polynomial level and are easily drowned out by noise; on the other hand, even if this does not affect the subsequent updates of $w_{j,m}$, latent tokens behave consistently with most $\boldsymbol{x}_j$ where $\mathbf{p}[j] \neq m$, failing to provide key information for $\hat{x}_m$. These factors limit the improvement of the model's reasoning capabilities. We validate the above theoretical conclusions through experiments and demonstrate the limitations of implicit reasoning chains in learning reasoning knowledge (see Figure 3).

### 4.2 LEARNING DIFFICULTY INCREASES EXPONENTIALLY WITH THE NUMBER OF COMPRESSED STEPS

To further consider the impact of implicit reasoning chains on the model's reasoning capabilities, in this section, we further analyze the learning difficulty of reasoning nodes in implicit reasoning chains. According to Theorem 1, the presence of child nodes for $S_e$ nodes makes their learning difficulty similar to that of explicit reasoning chains. Therefore, we focus on the learning difficulty of $S_d$ nodes.

Based on Theorem 2 from Kim & Suzuki (2025), Let $\mathbf{x}_m$ denote a node whose learning difficulty is measured by its average gradient. In gradient-based methods, larger gradients correspond to higher learning costs. For the parity problem, suppose the final output is learned directly without intermediate steps. Let $\widetilde{\nabla}$ be an $\varepsilon$-approximate gradient oracle that, with probability $1 - e^{-\Omega(d)}$ under random sampling, returns estimates of $\nabla L_n$. Then any iterative (possibly randomized) algorithm $\mathcal{A}$

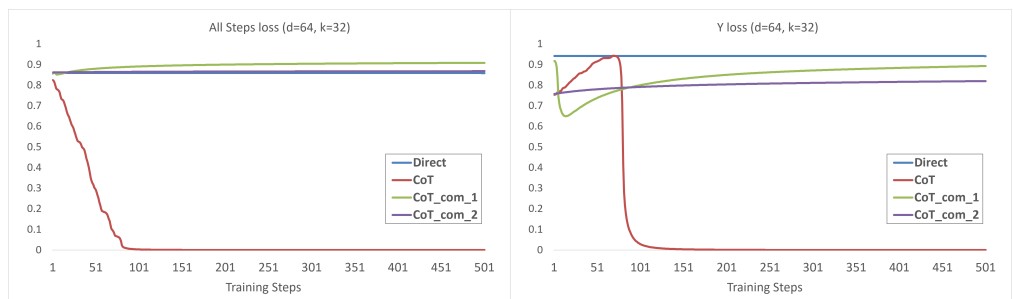

Figure 3: Experimental validation of Theorem 2. Experimental results on the 64-bit parity problem (training a single-layer Transformer, see Section 3.3 for architecture): The left/right subfigures show the training loss curves for all reasoning steps of the model as well as the final step, respectively. Compared to explicit CoT, implicit CoT fails to effectively learn the reasoning process (loss remains high and declines slowly).

making at most $T$ queries to $\widetilde{\nabla} L_n$ produces an output $\theta(\mathcal{A})$ whose $L_2$-loss satisfies the following lower bound:

$$\mathbb{E}_{p \in P, \boldsymbol{x}} \left[ (p(\boldsymbol{x}) - f_{\theta(\mathcal{A})}(\boldsymbol{x}))^2 \right] \geq 1 - \frac{4T}{\varepsilon^2} \left( \frac{1}{|P_k|} \vee \sqrt{\frac{4d}{n}} \right) \sup_{\theta, \boldsymbol{x}} \|\nabla f_\theta(\boldsymbol{x})\|^2 - 2e^{-\Omega(k)} \quad (7)$$

When evaluating the learning difficulty of $S_d$ nodes in implicit reasoning chains, the first $s$ steps are hidden, forcing the model to directly solve a $k = 2^s$-bit parity problem. Consequently, when examining how the number of compressed steps $s$ increases, we focus on the variation of $\mathbb{E}_{p \in P, \mathbf{x}} \left[ (p(\mathbf{x}) - f_{\theta(\mathcal{A})}(\mathbf{x}))^2 \right]$ with respect to $s$. For convenience, we express this dependence directly in terms of $k$. Then we have the following conclusion:

**Theorem 3.** *Suppose $n = e^{\Omega(k)}$ and $f_\theta$ has polynomially bounded gradients. Then there exists an $e^{-\Omega(k)}$-approximate gradient oracle $\widetilde{\nabla}$ such that, with probability at least $1 - e^{-\Omega(d)}$ over random sampling, the output $\theta(\mathcal{A})$ of any iterative (possibly randomized) algorithm making at most $O(\mathrm{poly}(k))$ queries to $\widetilde{\nabla} L_n$ satisfies the $L_2$-loss lower bound*

$$\mathbb{E}_{p \in P, \boldsymbol{x}} \left[ (p(\boldsymbol{x}) - f_{\theta(\mathcal{A})}(\boldsymbol{x}))^2 \right] \geq 1 - e^{-\Omega(k)}.$$

Theorem 3 shows that as the number of compressed steps increases, the model's learning task escalates from solving a 2-bit parity problem under full CoT to a $k$-bit parity problem. Because $k$ grows exponentially with the number of compressed steps $s$ in implicit reasoning chains, the learning difficulty of $S_d$ nodes likewise increases exponentially with $s$. To verify this empirically, we train implicit reasoning-chain models based on GPT-2 with varying numbers of compressed steps $s$ and record their test accuracy. Unlike Section 4.1, here we use the full GPT-2 model and prepend special tokens to switch between implicit and explicit reasoning, following prior work (Zhang et al., 2025a; Shen et al., 2025). Detailed experimental settings are provided in Appendix D. As shown in Figure 4, the learning cost of implicit reasoning chains rises sharply with $s$, exceeding even the case without CoT when all steps are compressed.

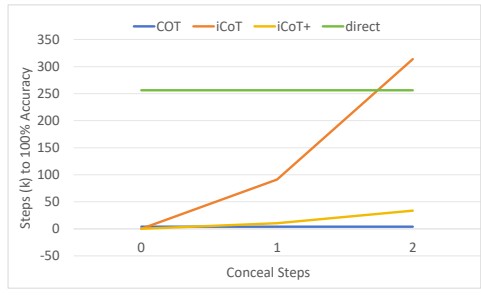

Figure 4: Learning difficulty rises exponentially with more compressed steps, but our iCoT+ method enables implicit reasoning chains to learn the reasoning process effectively.

# 5 EFFECTIVE SUPERVISION IS KEY FOR IMPLICIT COT TO LEARN REASONING KNOWLEDGE

Building on the above analysis, we systematically examine the limitations of implicit reasoning chains in acquiring reasoning knowledge. This naturally raises a question: how can their learning ability be enhanced? A key factor is the provision of effective supervision signals during backpropagation, enabling implicit reasoning chains to learn the hidden steps. However, due to the high-dimensional nature of latent tokens, they are far harder to supervise than discrete tokens. Consequently, most existing approaches provide little or no effective supervision (Zhang et al., 2025a; Shen et al., 2025), instead relying on latent tokens to spontaneously acquire reasoning patterns within their contexts. This limitation constrains the reasoning capacity of implicit reasoning chains (Zhu et al., 2025a). To address this, we propose a simple yet effective strategy: encouraging latent tokens to differentiate among reasoning steps through their spatial distributions.

As illustrated in Figure 5, a natural approach is to construct a decoder $f^\times$ that maps each implicit token to its corresponding reasoning-step labels: $f_c^\times(\mathbf{c}_s) = (x_{d+\tau_{s-1}+1}, \ldots, x_{d+\tau_s})$. This design indirectly conveys information about the compressed reasoning steps to the latent tokens, enabling them to acquire the reasoning associated with those steps within their context. For a complete reasoning step $\mathbf{s} = (x_1^s, \ldots, x_{\tau_s}^s)$, there are $2^{\tau_s}$ possible combinations, which requires the implicit token $\mathbf{c}_s$ to have sufficient expressive capacity to distinguish among them. Because different steps may encode different amounts of information, each step $s$ is assigned its own decoder $f_s^\times$. In practice, we represent all reasoning steps using a single implicit token to encourage effective compression of the reasoning chain.

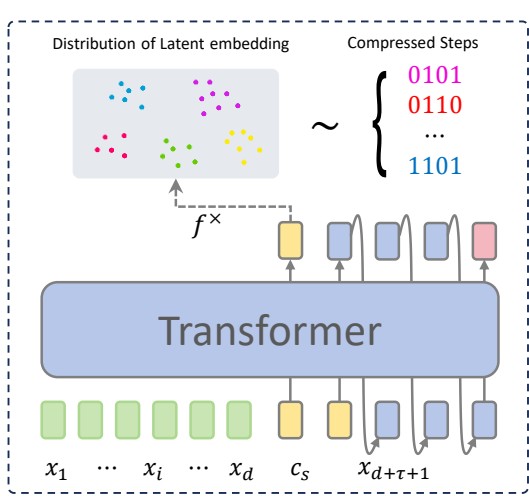

Figure 5: We construct a decoder $f^\times$ to control the distribution of latent tokens, enabling them to distinguish different reasoning steps.

In summary, our improvement method can be formalized by adding an additional supervision term to the training objective of the implicit reasoning chain:

$$L_n^{\text{iCoT+}}(\mathbf{W}) = (1-\lambda)L_n^{\text{iCoT}}(\mathbf{W}) + \lambda\frac{1}{n}\sum_{s=1}^{k-1}\sum_{i=1}^{n}\mathcal{L}(f_s^\times(\mathbf{c}^{s,i}), (x_{d+\tau_{s-1}+1}^i, \cdots, x_{d+\tau_s}^i)) \quad (8)$$

where $\mathcal{L}$ is a loss function measuring the discrepancy between predicted and true steps. For the parity problem, we use a multi-classifier to control the distribution, and $\lambda \in [0,1]$ is a hyperparameter that balances the weights of the two loss functions.

To evaluate the effectiveness of our proposed improvement, we adopt the experimental setup described in Section 4.2 for direct comparison. Prior work (Wies et al., 2023) shows that, even without CoT, Transformers can solve the parity problem but require substantially more training time. We therefore include both CoT-based and non-CoT methods as baselines. Specifically, we compare our enhanced approach (iCoT+) with the original implicit reasoning chain method (iCoT) under varying numbers of compressed steps. Learning efficiency is assessed by the number of training steps needed to reach 100% test accuracy. The results, presented in Figure 4, demonstrate that as the number of compressed steps increases, the learning difficulty of implicit reasoning chains grows exponentially—eventually exceeding that of non-reasoning-chain methods when all steps are compressed. In contrast, by aligning distributions, implicit reasoning chains can more effectively acquire the reasoning process, markedly reducing their learning difficulty.

# 6    CAN DISTRIBUTION ALIGNMENT ENHANCE REASONING CAPABILITIES?

Beyond improving the learning efficiency of implicit reasoning chains, it is also essential to assess whether distribution alignment enhances their reasoning capacity. To this end, we evaluate on the GSM8K dataset (Cobbe et al., 2021), which provides high-quality reasoning chains as supervision. This benchmark not only represents a reasoning-intensive mathematical task but also tests the effectiveness of reasoning-chain compression in less structured and more ambiguous natural-language reasoning scenarios.

We conduct experiments with the GPT-2 series models, which were pretrained before the release of GSM8K and thus avoid potential data leakage. Owing to their relatively small size, however, these models struggle to perform mathematical reasoning on GSM8K even with CoT. To mitigate this, we relax the reasoning phase by allowing the model to infer the final answer given the latent tokens and the remaining intermediate steps.

We train models using CoT, iCoT, and iCoT+, and compare their test accuracy under varying numbers of compressed steps $s$. Additional experimental details are provided in Appendix E. As shown in table 1, increasing $s$ gradually degrades the reasoning ability of implicit reasoning chains, whereas distribution alignment markedly enhances it. Notably, the advantage of iCoT+ diminishes at larger $s$, which we attribute to the growing difficulty of capturing intermediate-step information as more steps are compressed, thereby limiting further improvements in reasoning capacity.

| | | GPT2(124M) | | GPT2-Large(774M) | | GPT2-Xl(1.5B) | |
| --- | --- | --- | --- | --- | --- | --- | --- |
| | | accuracy | con-steps | accuracy | con-steps | accuracy | con-steps |
| full steps CoT | | 86.02 | 120 | 88.90 | 40 | 95.37 | 50 |
| conceal 1 step | icot | 83.70 | **100** | 89.30 | 40 | 95.83 | 50 |
| | icot+ | **86.15** | 180 | **91.20** | 40 | **96.82** | 50 |
| conceal 2 steps | icot | 64.56 | **100** | 67.80 | 40 | 71.28 | 50 |
| | icot+ | **65.57** | 180 | **69.00** | 40 | **72.22** | 50 |

Table 1: We conduct mathematical reasoning experiments on models of different sizes. "accuracy" denotes the final accuracy of the model, while "con-steps" represents the number of training steps required for the model to reach 50% of its final accuracy, indicating the steps needed for rapid convergence.

Although iCoT+ achieves efficient learning on the parity problem, its convergence in GPT-2 mathematical reasoning lags behind other methods, and iCoT shows no marked decline in convergence as steps are further compressed. We attribute this to two factors: 1) the added distribution-alignment loss demands substantial internal adjustment in Transformer models pretrained on large autoregressive corpora, thereby slowing convergence; and 2) in natural-language reasoning, models often exploit token-level shortcuts rather than performing genuine reasoning, unlike the parity task, which forces learning of complex logical relations from limited token types. These observations, in line with Lin et al. (2025), stand in sharp contrast to larger models, whose reasoning appears to draw primarily on pre-existing latent capacities rather than patterns learned exclusively from training data.

# 7    CONCLUSION

This paper systematically analyzes the limitations of implicit reasoning chains in acquiring reasoning knowledge. We show that the absence of intermediate reasoning nodes impedes spontaneous learning, causing overall learning difficulty to grow exponentially as more steps are compressed. These factors constrain the model's efficiency on complex reasoning tasks. Motivated by these theoretical insights, we advocate incorporating supervision signals for latent tokens and propose a simple yet effective method that encourages them to differentiate reasoning steps via their spatial distributions. We validate our approach and theoretical findings on both synthetic parity datasets and mathematical reasoning tasks. The results highlight the critical role of distribution alignment in reasoning-intensive scenarios. Future work will explore more advanced alignment strategies to further enhance the reasoning capabilities of implicit reasoning chains.

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

## A    APPENDIX

## B    PROOF OF THEOREM 2

Thanks to the analysis of the parity problem in Kim & Suzuki (2025), we have:

**Lemma 4** (concentration of interaction terms). *If each bit $x_j^i$ for $i \in [n]$, $j \in [d]$ is i.i.d. generated from the uniform distribution on $\{\pm 1\}$, for any $p > 0$ it holds with probability at least $1 - p$ that*

$$\max_{\substack{1 \le r \le 4 \\ (j_1, \cdots, j_r) \in I_{r,m}}} \frac{|\langle \boldsymbol{x}_{j_1}, \cdots, \boldsymbol{x}_{j_r}\rangle|}{n} \le \kappa := \sqrt{\frac{2}{n} \log \frac{32 d^4}{p}}.$$

*Proof.* Each tuple $(j_1, \cdots, j_r) \in I_{r,m}$ computes a specific nontrivial parity $x_{j_1} \cdots x_{j_r}$ for which the bits $x_{j_1}^i \cdots x_{j_r}^i$, $i = 1, \cdots, n$ are i.i.d. $\mathrm{Unif}(\{\pm 1\})$ due to symmetry. By Hoeffding's inequality we have that

$$\Pr\left(|\langle \boldsymbol{x}_{j_1}, \cdots, \boldsymbol{x}_{j_r}\rangle| \ge \lambda\right) \le 2 e^{-\lambda^2/2n}.$$

Moreover, $|I_{r,m}| \le (d + k - 1)^r \le (2d - 1)^r$ so that

$$|I_{1,m}| + \cdots + |I_{4,m}| \le (2d - 1) + \cdots + (2d - 1)^4 < (2d)^4.$$

Therefore it follows by union bounding that

$$\Pr\left(\max_{1 \le r \le 4, (j_1, \cdots, j_r) \in I_{r,m}} |\langle \boldsymbol{x}_{j_1}, \cdots, \boldsymbol{x}_{j_r}\rangle| \ge \lambda\right) \le 32 d^4 e^{-\lambda^2/2n},$$

which implies the statement.

$\square$

In particular, we take $n = \Omega(d^{2+\epsilon})$ and $p = \exp(-d^{\epsilon/2})$ so that $\kappa = O(d^{-1-\epsilon/4})$. This will ensure that the informative gradient signals will dominate the irrelevant interaction terms.

We further consider the case when latent tokens are present. We initialize the implicit token values as $\boldsymbol{c}_s = (\upsilon)_{i=1}^n$, where $-1 \le \upsilon \le 1$, allowing $\boldsymbol{c}_s$ to update in both positive and negative directions during training. Based on Lemma 4, when some variables are latent tokens, we have:

$$\frac{1}{n}|\langle \boldsymbol{x}_{j_1}, \cdots, \boldsymbol{x}_{j_{r-1}}, \boldsymbol{c}_j\rangle| \le \frac{1}{n}|\langle \boldsymbol{x}_{j_1}, \cdots, \boldsymbol{x}_{j_{r-1}}, \boldsymbol{x}_j\rangle| = O(\kappa). \tag{9}$$

Furthermore, we use the simplified notation $(\boldsymbol{x}_1, \cdots, \boldsymbol{x}_{d+k-1})$ to represent the contextual sequence containing implicit reasoning chains $(\boldsymbol{x}_1, \cdots, \boldsymbol{x}_d, \boldsymbol{c}_1, \cdots, \boldsymbol{c}_\tau, \boldsymbol{x}_{d+\tau+1}, \cdots, \boldsymbol{x}_{d+k-1})$, and denote $\boldsymbol{x}_i = \boldsymbol{c}_i$ for $i = d+1, \cdots, d+\tau$. At this point, we have:

$$L(\mathbf{W}) = \frac{1}{2n} \sum_{j=d+\tau_s+1}^{d+k-1} \|\phi(\hat{\boldsymbol{z}}_m) - \boldsymbol{x}_m\|^2, \quad \hat{\boldsymbol{z}}_m = \sum_{j=1}^{m-1} \sigma_j(\boldsymbol{w}_m)\boldsymbol{x}_j.$$

It is straightforward to verify for $1 \le \alpha < m$ that

$$\frac{\partial \sigma_\alpha(\boldsymbol{w}_m)}{\partial w_{j,m}} = (\delta_{j\alpha} - \sigma_\alpha(\boldsymbol{w}_m))\sigma_j(\boldsymbol{w}_m) = (\delta_{j\alpha} - \sigma_j(\boldsymbol{w}_m))\sigma_\alpha(\boldsymbol{w}_m)$$

and

$$\frac{\partial \hat{\boldsymbol{z}}_m}{\partial w_{j,m}} = \sum_{\alpha=1}^{m-1} (\delta_{j\alpha} - \sigma_j(\boldsymbol{w}_m))\sigma_\alpha(\boldsymbol{w}_m)\boldsymbol{x}_\alpha = \sigma_j(\boldsymbol{w}_m)(\boldsymbol{x}_j - \hat{\boldsymbol{z}}_m).$$

Then the gradient of $L$ with respect to each element $w_{j,m}$ at initialization can be computed as

$$\frac{\partial L}{\partial w_{j,m}}(\mathbf{W}) = \frac{1}{n}(\phi(\hat{\boldsymbol{z}}_m) - \boldsymbol{x}_m)^\top \frac{\partial \phi(\hat{\boldsymbol{z}}_m)}{\partial w_{j,m}}$$

$$= \frac{\sigma_j(\boldsymbol{w}_m)}{n} \langle \phi(\hat{\boldsymbol{z}}_m) - \boldsymbol{x}_m, \phi'(\hat{\boldsymbol{z}}_m), \boldsymbol{x}_j - \hat{\boldsymbol{z}}_m \rangle \tag{10}$$

$$= -\frac{1}{n(m-1)} \langle \boldsymbol{x}_m, 2c\hat{\boldsymbol{z}}_m, \boldsymbol{x}_j - \hat{\boldsymbol{z}}_m \rangle \tag{11}$$

$$+ \frac{1}{n(m-1)} \langle -\mathbf{1}_n + c\hat{\boldsymbol{z}}_m^2, 2c\hat{\boldsymbol{z}}_m, \boldsymbol{x}_j - \hat{\boldsymbol{z}}_m \rangle \tag{12}$$

$$+ \frac{1}{n(m-1)} \langle O(|\hat{\boldsymbol{z}}_m|^4), 2c\hat{\boldsymbol{z}}_m, \boldsymbol{x}_j - \hat{\boldsymbol{z}}_m \rangle \tag{13}$$

$$+ \frac{1}{n(m-1)} \langle \phi(\hat{\boldsymbol{z}}_m) - \boldsymbol{x}_m, O(|\hat{\boldsymbol{z}}_m|^3), \boldsymbol{x}_j - \hat{\boldsymbol{z}}_m \rangle. \tag{14}$$

**Computing interaction strengths.** The term (11) will be shown to contain the dominating gradient signal when $j = \mathsf{c}_1[m], \mathsf{c}_2[m]$, while the other terms can be bounded as perturbations. Let $\ell = \mathsf{h}_2[m]$ so that $x_m$ computes a $2^\ell$-parity.

For term (11), we substitute $\hat{\boldsymbol{z}}_m = \frac{1}{m-1}\sum_\alpha \boldsymbol{x}_\alpha$ at initialization to expand

$$\frac{1}{n} \langle \boldsymbol{x}_m, \hat{\boldsymbol{z}}_m, \boldsymbol{x}_j - \hat{\boldsymbol{z}}_m \rangle = \frac{1}{n(m-1)} \sum_\alpha \langle \boldsymbol{x}_m, \boldsymbol{x}_\alpha, \boldsymbol{x}_j \rangle - \frac{1}{n(m-1)^2} \sum_{\alpha,\beta} \langle \boldsymbol{x}_m, \boldsymbol{x}_\alpha, \boldsymbol{x}_\beta \rangle,$$

where the dummy indices $\alpha, \beta, \cdots$ are taken to run over $[m-1]$. Let us evaluate the third-order interaction terms $\langle \boldsymbol{x}_m, \boldsymbol{x}_\alpha, \boldsymbol{x}_\beta \rangle$. If $\mathsf{h}[\alpha] = \ell$, $x_m x_\alpha$ computes the parity of $2^{\ell+1}$ independent bits from $x_1, \cdots, x_d$ so $x_m x_\alpha x_\beta$ cannot be trivial, hence $(m, \alpha, \beta) \in I_{3,m}$ and $|\langle \boldsymbol{x}_m, \boldsymbol{x}_\alpha, \boldsymbol{x}_\beta \rangle| \le n\kappa$ by Lemma 4. Similarly, $\mathsf{h}[\beta] = \ell$ implies that $(m, \alpha, \beta) \in I_{3,m}$. Suppose $\mathsf{h}[\alpha], \mathsf{h}[\beta] \le \ell - 1$; unless $\mathsf{h}[\alpha] = \mathsf{h}[\beta] = \ell - 1$, the combined parity $x_\alpha x_\beta$ will not contain enough independent bits to cancel out the $2^\ell$ bits in $x_m$, so again $(m, \alpha, \beta) \in I_{3,m}$. Moreover if $\mathsf{h}[\alpha] = \mathsf{h}[\beta] = \ell - 1$, $x_m x_\alpha x_\beta$ will be trivial if and only if $\{\alpha, \beta\} = \{\mathsf{c}_1[m], \mathsf{c}_2[m]\}$, in which case $\langle \boldsymbol{x}_m, \boldsymbol{x}_\alpha, \boldsymbol{x}_\beta \rangle = n$. Thus we have that

$$\frac{1}{n} \sum_\alpha \langle \mathbf{x}_m, \mathbf{x}_\alpha, \mathbf{x}_\beta \rangle = \frac{1}{n} \sum_{\alpha \in S_i, \beta \notin S_i} \langle \mathbf{x}_m, \mathbf{x}_\alpha, \mathbf{x}_\beta \rangle + \frac{1}{n} \sum_{\alpha \notin S_i, \beta \in S_i} \langle \mathbf{x}_m, \mathbf{x}_\alpha, \mathbf{x}_\beta \rangle$$

$$+ \frac{1}{n} \sum_{(\alpha,\beta) \in S_i} \langle \mathbf{x}_m, \mathbf{x}_\alpha, \mathbf{x}_\beta \rangle + \frac{1}{n} \sum_{(\alpha,\beta) \notin S_i} \langle \mathbf{x}_m, \mathbf{x}_\alpha, \mathbf{x}_\beta \rangle$$

It is worth noting that when $m \in S_e$, since all its child nodes are not omitted, we have:

$$\frac{1}{n} \sum_{(\alpha,\beta) \notin S_i} \langle \mathbf{x}_m, \mathbf{x}_\alpha, \mathbf{x}_\beta \rangle = 2_{\{m \in S_e\}} + \frac{1}{n} \sum_{(m,\alpha,\beta) \in I_{3,m}} \langle \boldsymbol{x}_m, \boldsymbol{x}_\alpha, \boldsymbol{x}_\beta \rangle$$

$$= 2_{\{m \in S_e\}} + O((m-1)^2 \kappa).$$

$$\frac{1}{n} \sum_{(\alpha,\beta) \in S_i} \langle \mathbf{x}_m, \mathbf{x}_\alpha, \mathbf{x}_\beta \rangle = v^2 O((\tau_s)^2 \kappa) \leq O((m-1)^2 \kappa)$$

$$\frac{1}{n} \sum_{\alpha \in S_i, \beta \notin S_i} \langle \mathbf{x}_m, \mathbf{x}_\alpha, \mathbf{x}_\beta \rangle + \frac{1}{n} \sum_{\alpha \notin S_i, \beta \in S_i} \langle \mathbf{x}_m, \mathbf{x}_\alpha, \mathbf{x}_\beta \rangle = O((m-1)^2 \kappa)$$

So that

$$\frac{1}{n} \sum_\alpha \langle \mathbf{x}_m, \mathbf{x}_\alpha, \mathbf{x}_\beta \rangle = O((m-1)^2 \kappa) + O((m-1)^2 \kappa) + 2_{\{m \in S_e\}} + O((m-1)^2 \kappa)$$

$$= 2_{\{m \in S_e\}} + O((m-1)^2 \kappa).$$

Similarly, for the contraction $\langle \boldsymbol{x}_m, \boldsymbol{x}_\alpha, \boldsymbol{x}_j \rangle$, since $\boldsymbol{x}_j$ does not form a trivial with any stage, and $\boldsymbol{x}_m$ and $\boldsymbol{x}_\alpha$ cannot form a trivial, according to equation **??**, we have:

$$\frac{1}{n} \sum_\alpha \langle \boldsymbol{x}_m, \boldsymbol{x}_\alpha, \boldsymbol{x}_j \rangle = O((m-1)\kappa)$$

Since $\kappa = O(d^{-1-\epsilon/4})$ and $d < m \leq 2d - 1$, we can therefore isolate the leading term of order $\Theta(d^{-2})$ as

$$-\frac{1}{n(m-1)} \langle \boldsymbol{x}_m, 2c\hat{\boldsymbol{z}}_m, \boldsymbol{x}_j - \hat{\boldsymbol{z}}_m \rangle$$

$$= -\frac{2c}{(m-1)^2} O(d\kappa) + \frac{2c}{(m-1)^3} \left( 2_{\{m \in S_e\}} + O((m-1)^2 \kappa) \right)$$

$$= O(d^{-2-\epsilon/4}).$$

Next, for term (12), we expand

$$\frac{1}{n} \left\langle -\mathbf{1}_n + c\hat{\boldsymbol{z}}_m^2, 2c\hat{\boldsymbol{z}}_m, \boldsymbol{x}_j - \hat{\boldsymbol{z}}_m \right\rangle = -\frac{2c}{n} \langle \hat{\boldsymbol{z}}_m, \boldsymbol{x}_j \rangle + \frac{2c}{n} \langle \hat{\boldsymbol{z}}_m^2 \rangle + \frac{2c^2}{n} \langle \hat{\boldsymbol{z}}_m^3, \boldsymbol{x}_j \rangle - \frac{2c^2}{n} \langle \hat{\boldsymbol{z}}_m^4 \rangle.$$

The second-order terms can be computed as

$$\frac{1}{n}\left\langle \hat{\boldsymbol{z}}_m, \boldsymbol{x}_j\right\rangle = \frac{1}{n(m-1)}\left(\sum_{\alpha\in S_i}\left\langle\boldsymbol{x}_\alpha,\boldsymbol{x}_j\right\rangle + \sum_{\alpha\notin S_i}\left\langle\boldsymbol{x}_\alpha,\boldsymbol{x}_j\right\rangle\right)$$

$$= \frac{\tau_s}{m-1}\upsilon^2 + \frac{m-\tau_s}{m-1}O(\kappa) \le \frac{\tau_s}{m-1}\upsilon^2 + O(\kappa)$$

$$\frac{1}{n}\left\langle\hat{\boldsymbol{z}}_m^2\right\rangle = \frac{1}{n(m-1)^2}\left(\sum_{(\alpha,\beta)\notin S_i}\left\langle\boldsymbol{x}_\alpha,\boldsymbol{x}_\beta\right\rangle + \sum_{(\alpha,\beta)\in S_i}\left\langle\boldsymbol{x}_\alpha,\boldsymbol{x}_\beta\right\rangle\right.$$

$$\left.+ \sum_{\alpha\in S_i,\beta\notin S_i}\left\langle\boldsymbol{x}_\alpha,\boldsymbol{x}_\beta\right\rangle + \sum_{\alpha\notin S_i,\beta\in S_i}\left\langle\boldsymbol{x}_\alpha,\boldsymbol{x}_\beta\right\rangle\right)$$

$$= \frac{1}{n(m-1)^2}\left(nO((m-1)^2\kappa) + n\tau_s^2\upsilon^2 + 2n\upsilon O((m-1)^2\kappa)\right)$$

$$= \frac{\tau_s^2\upsilon^2}{(m-1)^2} + O(\kappa) \le \frac{\tau_s}{(m-1)}\upsilon^2 + O(\kappa)$$

Before discussing the fourth-order term, it is known that in the absence of latent tokens, $\frac{1}{n}\left\langle\hat{\boldsymbol{z}}_m^4\right\rangle = \frac{|[m-1]^4\setminus I_{4,m}|}{(m-1)^4} + \frac{|I_{4,m}|}{(m-1)^4}$ (Kim & Suzuki, 2025). According to 9, it is clear that in the presence of latent tokens, we have:

$$\frac{1}{n}\left\langle\hat{\boldsymbol{z}}_m^4\right\rangle \le \frac{|[m-1]^4\setminus I_{4,m}|}{(m-1)^4} + \frac{|I_{4,m}|}{(m-1)^4} = O(d^{-2}+\kappa). \tag{15}$$

We verify the calculation of the fourth-order term. We analyze by examining the cases of trivial terms of different orders. Without loss of generality, suppose $\alpha\le\beta\le\gamma\le\delta$.

(i) For $(\alpha,\beta)\notin I_{2,m}$, it is clear that in the parity calculation, only the case $\alpha=\beta$ exists, so there are at most $O(d)$ such cases.

(ii) For $(\alpha,\beta,\gamma)\notin I_{3,m}$, the parity calculation will be trivial only when $\mathsf{h}[\alpha]=\mathsf{h}[\beta]=\mathsf{h}[\gamma]$, so there are at most $O(d)$ such cases.

(iii) For $(\alpha,\beta,\gamma,\delta)\notin I_{4,m}$, according to the analysis of Kim & Suzuki (2025), there are a total of $O(d^2)$ such cases.

Hence it follows that

$$\frac{1}{n}\left\langle\hat{\boldsymbol{z}}_m^4\right\rangle \le \frac{1}{n(m-1)^4}\sum_{j=0}^{4}\left(\binom{4}{j}\sum_{\beta_1,\cdots,\beta_{4-q}}\left\langle\boldsymbol{x}_{\beta_1},\cdots,\boldsymbol{x}_{\beta_{4-q}}\right\rangle\right)$$

$$= \frac{|[m-1]^4\setminus I_{4,m}|}{(m-1)^4} + \frac{|I_{4,m}|}{(m-1)^4}O(\kappa) = O(d^{-2}+\kappa).$$

Furthermore, we consider the case of $(\alpha,\beta,\gamma)\notin I_{3,m}$, there are at most $O(d)$ such cases. Hence we also have:

$$\frac{1}{n}\left\langle\hat{\boldsymbol{z}}_m^3,\boldsymbol{x}_j\right\rangle = \frac{1}{n(m-1)^3}\upsilon\sum_{\alpha,\beta,\gamma}\left\langle\boldsymbol{x}_\alpha,\boldsymbol{x}_\beta,\boldsymbol{x}_\gamma\right\rangle = \frac{O(d)}{(m-1)^3} + O(\kappa) = O(d^{-2}+\kappa).$$

Combining the above, we obtain that

$$\frac{1}{n(m-1)}\left\langle -\boldsymbol{1}_n + c\hat{\boldsymbol{z}}_m^2, 2c\hat{\boldsymbol{z}}_m, \boldsymbol{x}_j - \hat{\boldsymbol{z}}_m\right\rangle = -\frac{2c\tau_s\upsilon^2}{(m-1)^2} + \frac{2c\tau_s\upsilon^2}{(m-1)^2} + \frac{O(\kappa)}{m-1}$$

$$= O(d^{-2-\epsilon/4}).$$

For term (13), we note that $\langle |\hat{\boldsymbol{z}}_m|^4 \rangle = \langle \hat{\boldsymbol{z}}_m^4 \rangle = O(nd^{-2} + n\kappa)$ as derived above. Then since each component of $\hat{\boldsymbol{z}}_m$, $\boldsymbol{x}_j - \hat{\boldsymbol{z}}_m$ are contained in $[-1,1], [-2,2]$, respectively, we have that

$$\frac{1}{n(m-1)} \left\langle O(|\hat{\boldsymbol{z}}_m|^4), 2c\hat{\boldsymbol{z}}_m, \boldsymbol{x}_j - \hat{\boldsymbol{z}}_m \right\rangle = \frac{4c}{n(m-1)} O(\langle |\hat{\boldsymbol{z}}_m|^4 \rangle) = O(d^{-2-\epsilon/4}).$$

Finally for term (14), by the Cauchy-Schwarz inequality we have

$$\frac{1}{n} \left\langle |\hat{\boldsymbol{z}}_m|^3 \right\rangle = \frac{1}{n} \sum_{i=1}^{n} |\hat{z}_{m,i}|^3$$

$$\leq \frac{1}{n} \left( \sum_{i=1}^{n} \hat{z}_{m,i}^2 \right)^{1/2} \left( \sum_{i=1}^{n} \hat{z}_{m,i}^4 \right)^{1/2} = \frac{1}{n} \langle \hat{\boldsymbol{z}}_m^2 \rangle^{1/2} \langle \hat{\boldsymbol{z}}_m^4 \rangle^{1/2}$$

$$= \frac{1}{n} O(nd^{-1})^{1/2} \cdot O(nd^{-2} + n\kappa)^{1/2} = O(d^{-1-\epsilon/8}),$$

and so we may bound

$$\frac{1}{n(m-1)} \left\langle \phi(\hat{\boldsymbol{z}}_m) - \boldsymbol{x}_m, O(|\hat{\boldsymbol{z}}_m|^3), \boldsymbol{x}_j - \hat{\boldsymbol{z}}_m \right\rangle = \frac{4}{n(m-1)} O(\langle |\hat{\boldsymbol{z}}_m|^3 \rangle) = O(d^{-2-\epsilon/8}).$$

From (11)-(14) we conclude that

$$\frac{\partial L}{\partial w_{j,m}}(\mathbf{W}) = -\frac{2c}{(m-1)^2} 1_{\{\mathtt{p}[j]=m\}} + O(d^{-2-\epsilon/8}),$$

which shows that at initialization, each non-child node and each implicit token receives a negligible gradient signal of order $O(d^{-2-\epsilon/8})$, while each child node receives a strong gradient signal of order $\Theta(d^{-2})$. This is because latent tokens do not have direct relationships with any nodes, nor do they form trivial combinations with any other nodes.

$\square$

## C  PROOF OF THEOREM 3

Based on the analysis of Kim & Suzuki (2025), Let $\mathbf{x}_m$ denote a node whose learning difficulty is measured by its average gradient. In gradient-based methods, larger gradients correspond to higher learning costs. For the parity problem, suppose the final output is learned directly without intermediate steps. Let $\widetilde{\nabla}$ be an $\varepsilon$-approximate gradient oracle that, with probability $1 - e^{-\Omega(d)}$ under random sampling, returns estimates of $\nabla L_n$. Then any iterative (possibly randomized) algorithm $\mathcal{A}$ making at most $T$ queries to $\widetilde{\nabla}L_n$ produces an output $\theta(\mathcal{A})$ whose $L_2$-loss satisfies the following lower bound:

$$\mathbb{E}_{p \in P, \boldsymbol{x}} \left[ (p(\boldsymbol{x}) - f_{\theta(\mathcal{A})}(\boldsymbol{x}))^2 \right] \geq 1 - \frac{4T}{\varepsilon^2} \left( \frac{1}{|P_k|} \vee \sqrt{\frac{4d}{n}} \right) \sup_{\theta,\boldsymbol{x}} \|\nabla f_\theta(\boldsymbol{x})\|^2 - 2e^{-\Omega(k)}, \quad (16)$$

**Lemma 5.** *Consider the parity problem described in Section 3.1. For a ground set of size $d$, the total number of possible subsets involved in the $k$-parity problem ($k \leq d/2$) is*

$$|P_k| = \binom{d}{k} = O(e^k)$$

*Proof.* According to the definition of combinations, we have:

$$|P_k| = \binom{d}{k} = \frac{d!}{k!(d-k)!} = \frac{d(d-1)\cdots(d-k+1)}{k!} \geq \left(\frac{d}{k}\right)^k$$

Based on Stirling's formula, we have $k! \geq \left(\frac{k}{e}\right)^k$ ., therefore

$$\frac{d^k}{k!} \leq d^k \left(\frac{e}{k}\right)^k = \left(\frac{ed}{k}\right)^k.$$

So that $|P_k| = O(e^k)$.

$\square$

## D  EXPERIMENT DETAILS OF SECTION 4.2 AND 5

In these two experiments, we use GPT2 as the base model and set $d = 16$ and $k = 8$ as shown in Figure 2. With this setting, inferring the final output requires three steps. Consistent with Zhang et al. (2025a); Shen et al. (2025), we add special tokens $< com >$ and $< sep >$ to mark the beginning and end of implicit reasoning before and after the model performs implicit reasoning. That is, for each sample, the input is:

$$\underbrace{x_1, x_2, \cdots, x_d}_{\text{input tokens}}, \text{<com>}, \underbrace{c_1, c_2, \cdots, c_\tau}_{\text{implicit reasoning tokens}}, \text{<sep>}, \underbrace{x_{d+1}, x_{d+2}, \cdots, x_{d+k-1}}_{\text{explicit reasoning tokens}}.$$

To effectively supervise the learning of implicit chain-of-thought, we add a supervision signal at the last token of the implicit chain-of-thought to guide the model to learn the correct implicit chain-of-thought. Specifically, for the set of latent tokens $c^s$, we pass their output vectors through a multi-layer perceptron (MLP) for multi-class classification to predict the $2^\tau$ different possibilities of the compressed steps, thereby aligning the distribution of latent tokens with that of the correct implicit chain-of-thought. For explicit chain-of-thought, we directly use the cross-entropy loss function to supervise its learning.

We use the Adam optimizer with a learning rate of $1e - 6$. We train the model until it achieves $100\%$ accuracy on the validation set. For different loss functions, we record the number of steps required for training as a quantitative assessment of their learning difficulty.

## E  EXPERIMENT DETAILS OF SECTION 6

In this experiment, since previous works did not release their data and code, we used different models(Llama, GPT2) for instruction fine-tuning based on the llamafactory framework (Zheng et al., 2024). We modified the model and data loading parts to suit our experimental needs. A sample from GSM8K is as follows:

> **Question:** A robe takes 2 bolts of blue fiber and half that much white fiber. How many bolts in total does it take?
> **Answer:** It takes 2/2=«2/2=1»1 bolt of white fiber \n So the total amount of fabric is 2+1=«2+1=3»3 bolts of fabric \n #### 3.

For implicit chain-of-thought, we split the *Answer* by "\n" and use the results as each reasoning step of the chain-of-thought. When we compress the reasoning steps, we add a special token sequence [<com> <pause> $\cdots$ <pause> <sep>] before the answer to mark the implicit reasoning process. When we use iCoT+ for distribution alignment, we encode the reasoning steps represented by <pause> and use the mean squared error loss function to align their distribution with that of the complete chain-of-thought. Our code will be released later.

