# OpenReview forum: "Dissecting Implicit Chain of Thought: Can Transformers Learn It Spontaneously?"
_ICLR.cc/2026/Conference — Submitted to ICLR 2026_

### Official Review · Reviewer_2gG3 · 2025-10-30

**Soundness:** 2
**Presentation:** 3
**Contribution:** 2
**Rating:** 2
**Confidence:** 4

**Summary:**

The authors show that while compressing reasoning steps into latent tokens can reduce computational cost, it causes severe degradation in reasoning performance. Using theoretical analysis on the parity problem and Transformer models, the paper finds that latent tokens struggle to capture intermediate reasoning, and learning difficulty grows exponentially as more steps are compressed. To mitigate this, the authors propose a “distribution alignment” method that adds weak supervision to latent tokens, improving learning efficiency and reasoning stability. Experiments on parity and GSM8K show moderate improvements, especially for smaller compression levels.

**Strengths:**

The theoretical framework is carefully developed, connecting gradient decay and reasoning inefficiency. The introduction of the distribution alignment approach is conceptually simple and empirically validated, showing that partial supervision can help implicit CoT models retain reasoning ability. The study also bridges synthetic and natural reasoning tasks, which strengthens the relevance of the findings beyond toy problems.

**Weaknesses:**

the practical impact of the paper is limited. The parity problem setting, while analytically neat, is too artificial to convincingly generalize to real-world reasoning tasks. The paper’s claim that implicit CoT suffers exponential degradation is largely shown through controlled setups rather than realistic language modeling conditions. The proposed “distribution alignment” method, although improving accuracy slightly, is underspecified and loosely motivated—its mechanism for improving latent token expressivity is unclear and seems ad hoc.

The experiments on GSM8K use small GPT-2 models where reasoning is already weak, making the improvements less meaningful. Furthermore, the paper fails to provide ablations that clarify whether the observed benefits come from the supervision term itself or other training artifacts. The analysis section is mathematically heavy but lacks intuitive interpretation or visualization that connects to reasoning in language models.

the paper’s scope is narrow: it isolates implicit CoT without addressing how it interacts with real prompt-based reasoning or large-scale pretraining objectives, which weakens its practical relevance for understanding reasoning in modern LLMs.

**Questions:**

see above

---

### Official Review · Reviewer_4KYd · 2025-10-30

**Soundness:** 2
**Presentation:** 2
**Contribution:** 2
**Rating:** 4
**Confidence:** 3

**Summary:**

This paper tackles the challenge of declining performance in latent Chain-of-Thought (CoT) reasoning by offering a theoretical analysis of its training costs. The authors reformulate latent CoT reasoning as a $k$-parity problem and argue that the lack of effective supervision for latent tokens results in gradients that fail to guide the model in acquiring reasoning capabilities. As more reasoning steps are compressed into latent representations, the difficulty of learning increases exponentially. To mitigate this, the paper introduces the iCoT+ strategy, which encourages latent tokens to spatially differentiate among reasoning steps. Experimental results on the GPT-2 series models suggest improved performance.

Overall, this paper is somewhat an interesting investigation into the latent CoT reasoning. However, the theoretical analysis lacks a compelling motivation, and the experimental validation is insufficient.

**Strengths:**

1.  Reformulating implicit CoT reasoning as a $k$-parity problem is an interesting exploration. This provides insights into enhancing the performance of implicit CoT.

2. The proposed distribution alignment method for training implicit CoT demonstrates its effectiveness.

**Weaknesses:**

1. The motivation for formulating implicit CoT reasoning as the $k$-parity problem is unclear. The connection between the traditional parity problem and implicit CoT lacks a coherent conceptual transition, making the formulation difficult to follow. Additionally, the role of Section 3.3 is ambiguous.

2. The theoretical analysis of training implicit CoT relies heavily on [1]. This limits the originality and theoretical contribution of this work.

3. While the authors attempt to perform theoretical analyses to examine the limitations of implicit reasoning chains, it's unclear why these analyses can derive the proposed distribution alignment method.

4. Experiments are not solid. The results are limited to the GPT-2 series, without evaluation on modern LLMs. This narrow scope weakens the empirical support for the proposed iCoT+ strategy and raises concerns about its generalization ability.

[1] Kim et al., Transformers Provably Solve Parity Efficiently with Chain of Thought. ICLR'2025.

**Questions:**

1. Since the core idea of the distribution alignment is to differentiate the spatial distribution of latent tokens, how does this connect to the $k$-parity problem?
2. The details of the baseline method, iCoT, are not sufficiently described. How does it implement?
3. Beyond GPT-2, how does the proposed iCoT+ strategy perform on modern LLMs (e.g., Llama or Qwen series)?
4. The overall writing of this paper is not clear. I strongly suggest that the authors carefully improve the use of mathematical symbols and logical transitions across paragraphs.
5. Some typos need to be fixed, e.g., (1) in Figure 2, when s=2, "patent step" should be "Latent Step"; (2) In Line 201, $\mathbf{c}_s$ should be $\mathbf{c}^s$.

---

### Official Review · Reviewer_uuPU · 2025-10-31

**Soundness:** 2
**Presentation:** 2
**Contribution:** 2
**Rating:** 2
**Confidence:** 4

**Summary:**

This paper investigates the fundamental limitations of "Implicit Chain-of-Thought" (iCoT) in large language models.
The core of this paper is to theoretically analyze why iCoT fails and to propose a solution based on this theory. The authors prove that without effective supervision, latent tokens cannot spontaneously learn the intermediate reasoning processes. Furthermore, as more steps are compressed, the model's learning difficulty increases exponentially.
Based on this, the paper proposes a "distribution alignment method" called iCoT+. This method adds an auxiliary decoder during training, forcing the distribution of the latent tokens to align with the distribution of the true reasoning steps they represent, thereby providing the necessary supervision.
The paper validates its theory and method on the synthetic "Parity Problem" and the real-world "GSM8K" mathematical reasoning task. Experiments show that iCoT+ effectively improves the learning efficiency and final performance of iCoT, significantly mitigating the performance degradation caused by reasoning compression.

The main contributions of this paper are:
- Revealing the Theoretical Flaws of iCoT.
- Quantifying the Learning Difficulty of iCoT.
- Proposing the iCoT+ Solution (Distribution Alignment).

**Strengths:**

The paper demonstrates strong originality by providing a novel and formal theoretical analysis of a critical, empirically observed problem.

**Weaknesses:**

The experimental results on a single task (GSM8K) are not strong enough to fully support the paper's claims about improving modern LLM reasoning.
The experiments use the GPT-2 series (up to 1.5B parameters). These models are known to be poor mathematical reasoners, even with CoT, as the authors acknowledge. Findings on these models do not necessarily generalize to modern, reasoning-capable models.
Another significant weakness of the paper is the lack of precision and a somewhat superficial treatment of its core proposed method, "distribution alignment." This creates confusion for the reader and obscures the paper's technical contribution.

**Questions:**

Thank you for your work. The theoretical analysis of iCoT's limitations is insightful. However, the empirical validation of the proposed iCoT+ method feels preliminary. My main concern is the lack of supplementary experiments and the narrow scope of the models and datasets used, which makes it difficult to assess the method's robustness, generalizability, and practical utility.

A response to the following points would be very helpful in clarifying these issues and could significantly strengthen my view of the paper.

---

### 1. On the Lack of Supplementary Experiments (Ablations)

The paper introduces a new method (iCoT+) but omits several key ablation studies needed to understand how and why it works.

- **Q1 (Choice of Alignment Loss):**
  You implement your "distribution alignment method" in two different ways: using a *multi-classifier* for the parity problem and a *mean squared error loss function (MSE)* for GSM8K.
  - (a) Can you please justify this methodological inconsistency? Why was MSE chosen for the complex, high-dimensional embeddings of natural language steps?
  - (b) Did you test the classifier on GSM8K or the MSE loss on the parity task? Understanding this comparison is crucial to know if the choice of loss function is a critical design decision or an arbitrary one.

- **Q2 (Decoder Architecture):**
  The paper provides no details on the architecture of the auxiliary decoder.
  - (a) What architecture was used for the GSM8K experiments (e.g., a simple linear layer, a multi-layer MLP)?
  - (b) How does the performance of iCoT+ change with the complexity (e.g., number of layers/parameters) of this decoder? This is critical for understanding the method's trade-offs in terms of added computational overhead.

---

### 2. On the Limited Scope of Models and Datasets

The paper's practical claims currently rest on a single, outdated model family (GPT-2) and a single task (GSM8K). This narrow scope makes it difficult to assess the generalizability of your findings.

- **Q3 (Model Generalizability):**
  The experiments are conducted on the GPT-2 series. As you note, these models struggle with mathematical reasoning at baseline.
  - (a) How can we be confident that the benefits of iCoT+ are not just an artifact of fixing a deficiency in GPT-2's limited architecture?
  - (b) *Suggestion:* The paper's claims would be significantly strengthened if you could provide even preliminary results showing that iCoT+ also outperforms standard iCoT on a more modern, reasoning-capable open-source model (e.g., Llama 3 8B, Mistral 7B).

- **Q4 (Task Generalizability):**
  The validation is limited to one mathematical reasoning dataset (GSM8K).
  - (a) Do you have any results or insights on how iCoT+ performs on other types of reasoning tasks that also rely on CoT (e.g., logical reasoning like LogicBench, or commonsense reasoning like StrategyQA)?
  - (b) *Suggestion:* To make a more general claim about improving "reasoning chains," I would strongly recommend testing your method on at least one non-mathematical reasoning benchmark to demonstrate broader applicability.

---

If the additional experiments are sufficiently comprehensive, I would consider increasing my score.

---

### Official Review · Reviewer_KFAi · 2025-11-01

**Soundness:** 3
**Presentation:** 4
**Contribution:** 1
**Rating:** 4
**Confidence:** 3

**Summary:**

The paper studies the efforts to convert explicit CoT to implicit or latent CoT in LLMs. Existing approaches to do this are known to suffer from quality losses. The paper provides a theoretical backing for why this might be the case by showing for the problem of k-parity the training becomes harder when we move to a latent space as opposed to CoT. To remedy this, they propose a distribution alignment method for training implicit CoT.

The main results of the paper are:
1. They show that the gradients for implicit CoT are much smaller compared to that for explicit Cot for the parity problem.
2. The next Theorem shows that as the number of compressed steps increases the number of training steps required to learn the problem via gradient descent style methods increases exponentially with the number of compressed steps.
3. Distribution alignment method for training implicit CoT: The main motivation is to provide some form of partial supervision to the latent tokens. This can be done by mapping each implicit latent token to a set of explicit tokens that it is supposed to model via a decoder f^x. The authors propose using a different decoder for each implicit CoT step. They show an experiment that this improves the training convergence for the k-parity problem. They also experiment with this approach on GSM8k where they are able to show small improvements over a fully implicit CoT.

**Strengths:**

- The paper aims to theoretically understand a common empirical observation and provides a clean analysis using the problem of k-parity to demonstrate the difficulty of learning with implicit CoT

- The paper proposes an interesting distribution alignment approach to augment implicit CoT learning. However, not much empirical analysis is performed on the distribution alignment approach to show its efficacy across settings and across scales.

**Weaknesses:**

- My main concern is that the theoretical results in the paper seem to be reminiscent of the results of “How far can Transformers Reason? The Globally Barrier and the Inductive Scratchpad.” [Abbe, et. Al. 24]. A discussion of how the results of current paper compare to this paper is in order and as of now the older paper largely seems to subsume the conceptual theoretical message of the current work.

- The proposed distribution alignment solution is interesting but not enough experiments are done to fully vet it. The choices of d and k seem to be very small (d=16, k=8) and it is unclear how the empirical behavior would change if they were made higher. The experiments on GSM8k also show minimal improvements. In addition, due to using small sized models, the authors mention they ”relax the reasoning phase” to condition on remaining intermediate steps. Its unclear how this is done and how much impact this has on the strong numbers of iCoT in the table.

**Questions:**

- Can you shed more details on how the conditioning on the remaining intermediate steps is being done in Section 6?
- The theoretical results shown seem subsumed by prior work mentioned above. There might also be other related work to compare to such as “A Theory of Learning with Autoregressive Chain of Thought” [Joshi et. al. 25]. Can you elaborate on how your results differ in novelty from those in these prior works?

---

### Meta-Review · Area_Chair_V4YJ · 2026-01-12

**Summary:**

This paper studies implicit chain of thought (implicit CoT), where the CoT reasoning is trained to be internalized into the model, such that at inference time reasoning trace can be shortened. This work first analyzes the theoretical limitations of implicit CoT, and verifies it on a k-parity problem that the difficulty of learning to internalize intermediate reasoning steps grows exponentially with the number of steps internalized. Then this work proposes a method to improve the existing implicit CoT training approaches by supervising the internal hidden representations during internalization to reproduce the tokens being internalized. Experiments on grade school math reasoning found that this method outperforms the existing implicit CoT approach.

Reviewers overall share concerns on the experimental design: only a synthetic k-parity dataset and a GSM-8K dataset are used and results are not very significant. Also, some reviewers pointed out that there's an existing work ("How far can Transformers Reason? The Globally Barrier and the Inductive Scratchpad") that shows a similar analysis result.

Overall, I think this paper has great potential: the analysis on how the difficulty level of internalization grows with the number of steps being internalized is insightful and seems generally applicable to more tasks (for example, I recommend authors to check tasks used in similar works to implicit CoT such as searchformer). I think this paper can be greatly improved if authors conduct more extensive experiments on more commonly used datasets (and if resource permits, on bigger models) to show the generalization of the conclusions reached. However, due to the current limited experiments, I'm not recommending its acceptance in the current form. That said, I highly encourage authors to submit their next version with more extensive experiments to the next conference.

**Reviewer Concerns:**

No rebuttal submitted.

**Reviewer Scores:**

Since no rebuttal was submitted, I don't think reviewers would change their scores.

---

### Decision · Program_Chairs · 2026-01-26

Reject